# Impact of Fermentable Protein, by Feeding High Protein Diets, on Microbial Composition, Microbial Catabolic Activity, Gut Health and beyond in Pigs

**DOI:** 10.3390/microorganisms8111735

**Published:** 2020-11-05

**Authors:** Hanlu Zhang, Nikkie van der Wielen, Bart van der Hee, Junjun Wang, Wouter Hendriks, Myrthe Gilbert

**Affiliations:** 1Animal Nutrition Group, Department of Animal Sciences, Wageningen University, 338, 6700 AH Wageningen, The Netherlands; hanlu.zhang@wur.nl (H.Z.); nikkie.vanderwielen@wur.nl (N.v.d.W.); wouter.hendriks@wur.nl (W.H.); 2State Key Laboratory of Animal Nutrition, College of Animal Science and Technology, China Agricultural University, Beijing 100193, China; wangjj@cau.edu.cn; 3Division of Human Nutrition and Health, Department of Agrotechnology and Food Sciences, Wageningen University, Stippeneng 4, 6708 WE Wageningen, The Netherlands; 4Host-Microbe Interactomics Group, Department of Animal Sciences, Wageningen University, 338, 6700 AH Wageningen, The Netherlands; bart.vanderhee@wur.nl; 5Laboratory of Microbiology, Department of Agrotechnology and Food Sciences, Wageningen University, Stippeneng 4, 6708 WE Wageningen, The Netherlands

**Keywords:** protein fermentation, dietary protein, microbial composition, fermentation metabolites, gut health, pig

## Abstract

In pigs, high protein diets have been related to post-weaning diarrhoea, which may be due to the production of protein fermentation metabolites that were shown to have harmful effects on the intestinal epithelium in vitro. In this review, we discussed in vivo effects of protein fermentation on the microbial composition and their protein catabolic activity as well as gut and overall health. The reviewed studies applied different dietary protein levels, which was assumed to result in contrasting fermentable protein levels. A general shift to N-utilisation microbial community including potential pathogens was observed, although microbial richness and diversity were not altered in the majority of the studies. Increasing dietary protein levels resulted in higher protein catabolic activity as evidenced by increased concentration of several protein fermentation metabolites like biogenic amines in the digesta of pigs. Moreover, changes in intestinal morphology, permeability and pro-inflammatory cytokine concentrations were observed and diarrhoea incidence was increased. Nevertheless, higher body weight and average daily gain were observed upon increasing dietary protein level. In conclusion, increasing dietary protein resulted in higher proteolytic fermentation, altered microbial community and intestinal physiology. Supplementing diets with fermentable carbohydrates could be a promising strategy to counteract these effects and should be further investigated.

## 1. Introduction

Proteins, peptides and amino acids (AA) in the gastrointestinal tract of pigs, either from exogenous or endogenous origin, can be utilised by the inhabitant microbiota. This utilisation first requires the breakdown of larger proteins and peptides by microbiota-derived proteases and peptidases, so-called proteolytic activity [1]. Subsequently, AA and short peptides act as building blocks for microbial protein synthesis or they can be utilised as an energy source, often referred to as protein fermentation [2]. This dissimilatory metabolism is less energetically favourable compared to carbohydrate catabolism [3] and leads to a series of metabolites of which several have the potential to negatively affect the gut in vitro [4].

Protein fermentation is thought to occur mostly in the large intestine because of its greater microbial population and slower passage rate compared to the small intestine [5]. Dietary proteins, as well as endogenous proteinaceous material such as digestive enzymes, sloughed epithelial cells, mucins and microbes [6], reach the large intestine when not digested and absorbed in the upper gastrointestinal tract. Approximately 15 to 25% of the dietary proteins in a conventional pig diet reach the large intestine [7], but this can be greatly influenced by the dietary protein level and the digestibility of the included protein source [8,9]. Moreover, the fate of these proteins in the large intestine depends on the quantity and type of dietary fermentable carbohydrates present [10].

High protein diets have been related to decreased faecal consistency and increased incidence of post-weaning diarrhoea in pigs [11,12], with protein fermentation processes suspected as being the underlying cause [4]. The current review focuses on the effects of protein fermentation in vivo by comparing microbial composition, the formation of metabolites and gut health between pigs fed with increased dietary protein levels compared to pigs fed with lower protein levels.

## 2. Changes in Microbial Composition

The intestinal microbiota plays a major role in the modulation of host physiology and metabolism, including nutrient utilisation, bioavailability, energy status and immune system development [13,14]. After birth, the intestinal tract is rapidly colonised by microbiota and its composition changes over time in response to diet, stress and disease state [15,16]. Diet is a large driver of microbial composition in the intestine, especially at a young age, as bacterial composition is still developing and is neither stable nor resilient; microbiota composition in adult pigs is assumed to be relatively stable [17,18]. As such, weaning piglets are expected to be more prone to microbial changes in response to diet, with increased indigestible protein potentially related to unfavourable health outcomes.

Multiple studies have investigated the effects of dietary protein on microbial composition and host health [19,20,21]. However, there are several factors that make comparisons between such studies rather difficult. For example, the methodology used to analyse and report microbial composition varies, e.g., utilising different reference databases for operational taxonomic units (OTU) clustering, different sequencing techniques or genomic regions or depth of analysis. For instance, culture-based studies were only able to identify a few groups of known bacteria that were preselected by researchers. In the majority of these studies, total Coliforms and *E. coli* are chosen to represent potential harmful species and Lactobacilli are chosen to represent beneficial species [22,23,24]. Although culture-independent DNA sequencing methods have been developed [25], studies are still limited due to the required probes and incompleteness of the databases to analyse sequenced data. As not all microbes can be identified by OTU clustering methods due to similarity in sequenced gene regions, results on microbial composition and classification should be compared cautiously. Furthermore, sequencing of various regions of the hypervariable region sequencing (V) of the 16S SSU rRNA gene also affects the estimated results, where the V3/V4 region has shown the highest classification accuracy [26], but more recent studies signify the importance of utilising long-read sequencing of the full 16S gene to overcome inter- and intragenomic variation to more accurately estimate community profiles [27]. In addition, most studies calculate relative abundance from sequenced composition, but it is likely that microbial composition and biological implications are different when expressed in absolute abundance, rather than in relative abundance [28]. Other factors that may lead to differences in microbial composition results between studies [12,19,22] may be related to physiology such as intestinal segment and age or variation in other dietary components, especially fermentable carbohydrate. The microbial composition is known to vary between intestinal segments and factors such as pH and substrate availability in these segments play a role [29]. When increasing protein level, changes in microbial composition also showed segment-dependent changes across the jejunum, caecum, colon and in faeces. For instance, the proportions of major phyla were altered by protein level only in the ileum, but not in the colon [30]. In addition, the ileal microbiota structure can show different responses to an increase in dietary protein compared to the colon, especially at the family and genus level [31]. Despite the limitations for comparing studies, effects of dietary protein level on general trends in microbial composition can be summarised and are discussed below.

Microbial richness and diversity are important parameters in host–microbe symbiosis. Upon increasing crude protein level in the diet, the microbial richness and diversity, i.e., number and variety of OTUs, was found unaltered in most studies (Table 1). Nevertheless, some studies showed a temporary increase in microbial richness and diversity when providing higher dietary protein levels to pigs [31,32,33,34]. The richer and more diverse colonic microbiota of weanling piglets reported was possibly related to the enterotoxigenic *Escherichia coli* challenge and was only observed seven days post-challenge, although no effect on total Coliforms was observed in these piglets [32]. In general, higher microbiota richness and diversity is considered beneficial and protective [35,36]. However, this was not related to an advantageous health outcome and even worse faecal score was observed [32], indicating that such indices should be interpreted with caution. Interestingly, Peng et al. [34] showed a quadratic relationship between dietary protein level and microbial diversity in the colon, with the highest colonic microbial diversity with intermediate dietary protein level (15%). However, this quadratic relationship was not found in the ileum and caecum. Overall, microbial diversity and richness appear not to change much in response to increasing dietary protein level, although some studies found (non-linear) associations [31,32,33,34].

When focusing in more detail on microbial composition, complex and diverse responses to increased dietary protein were found in several studies (Table 1). For example, contrasting results can be found involving the abundance of Coliforms, which has been regarded as an indicator of the population of pathogens such as *Salmonella* species in pigs [37]. Wellock et al. [12] detected increased numbers of colonic and faecal Coliforms when feeding 18% dietary crude protein compared to 13%, whereas total Coliforms were not affected by dietary protein level in several other studies [19,22,32,38]. In addition, colonic Coliform numbers increased again while faecal Coliforms decreased when protein level was further increased to 23% compared 18% [12]. Therefore, changes in microbial composition upon increased dietary protein are also dose- and site-dependent.

At the phylum level, *Firmicutes* and *Bacteroidetes* account for the largest proportion of colonic microbiota and its species are mainly strict anaerobes [16,31]. Many species in these two phyla are nitrogen (N-)fermenting and can utilise peptides and amino acids through different catabolic pathways [39]. With increasing the dietary protein level, *Firmicutes* counts were increased in the caecum [33], whereas no effect on relative abundances or numbers of *Firmicutes* were found in the caecum, colon and faeces in other studies [40,41,42]. This difference between studies could be attributed to the longer experimental period in the latter studies, as the greater values of *Firmicutes* in the study of Lou et al. [33] was only detected on experimental day 25, while no differences were found later at day 45. *Bacteroidetes* counts are generally not affected by higher protein intake (Table 1), although reduced counts were observed in a study with antibiotic intervention in the early life [41]. Although early antibiotic intervention had minimal effect on the influence that dietary protein level had on microbial composition in the latter study, it could explain the difference with studies that do not show an effect of increasing dietary protein level on *Bacteroidetes* abundance [33,34] or proportion [30]. Overall, at the phylum level, also dose-dependent effects were shown [31], as an increase in the proportion of *Firmicutes* (and conversely reduced *Proteobacteria* or *Bacteroidetes*) in the ileum and colon was observed with 15% dietary protein compared to 12% and 18%. Similarly, increased *Firmicutes* and decreased *Proteobacteria* were detected when feeding a 16% dietary crude protein diet compared to feeding 10%, whereas no differences in phyla proportions in the ileum were found in the 13% dietary crude protein group compared to 10% [30].

Shifts in microbial composition at family and genus level have been observed in pigs fed different dietary protein levels (Table 1). For instance, increased populations of *Clostridium* and *Streptococcus* were found in pigs fed with higher dietary crude protein level [11,22,31], which may be explained by the fact that these are major AA-metabolising bacteria [2]. An increase in these groups has been associated with increased risk of infection and animal disease [43]. However, increasing crude protein level in diets for weanling pigs also increased the number of caecal and colonic Lactobacilli [12] and *Bifidobacterium* [34], which are generally considered to be beneficial bacteria that prevent pathogens from overgrowing in the intestine [12,31,44,45]. At the family and genus level, dose-dependent effects have also been found. For example, increasing crude dietary protein level from 14% to 20%, but not 17%, increased the relative abundance of *Lactobacillus, Turicibacter* and *Ruminococcus* and decreased the relative abundance of *Prevotella* and *Lachnospira* in the colon of growing pigs [46]. It has been suggested that a 3%-unit reduction of dietary protein may not be a sufficiently large contrast to shift the colonic microbiota composition, which, during later life and under low infection pressure shows high compositional stability and resilience [46]. In contrast, there are also studies showing that 3% units dietary protein restriction could alter the microbial composition. Compared to 10% dietary crude protein level, pigs fed 13% crude protein reduced the proportion of *Clostridium* and *Escherichia-Shigella,* as well as increasing the proportion of *Peptostreptococcaceae* in ileum and colon [30]. In addition to dose-dependent (linear effect), quadric relations between protein level and microbial composition were also found. For instance, pigs fed with a 15% dietary crude protein level showed the lowest proportion of *Streptococcaceae* and highest proportion of *Lactobacillaceae* in the ileum, as well as the lowest *Ruminococcaceae_UCG-005* and highest *Veillonellaceae* in the colon, compared to 12% or 18% dietary crude protein [31].

Microbial changes at species level are also summarised in Table 1. Most studies found increased *E. coli* counts [22,41,42], or no response in *E. coli* counts when feeding high protein levels [19,38]. On the other hand, colonic *E. coli* showed a quadric effect as it decreased when dietary crude protein level was increased from 14% to 15% but increased when protein level was further increased to 17.2% and 20% [34]. Among these studies, only one analysed the enterotoxigenic *E. coli* while many species counted in other studies could also include non-pathogenic *E. coli* types [22]. Further, no conclusive results on species such as *Clostridium* cluster IV and *Clostridium* cluster XIVa can be made. These butyrate-producing bacteria showed conflicting results in the caecum between two studies [33,41], with study duration potentially contributing to these differences.

Overall, it is difficult to pinpoint the precise changes in pig intestinal microbial composition with increasing dietary protein levels. In most of the studies, microbial diversity and richness were not affected, while increased populations of N-fermenting bacteria like *Clostridium* and *Streptococcus* were sometimes observed. Nevertheless, still many contrasting results were found, which are difficult to interpret due to differences in techniques and experimental design. To evaluate the potential effects of dietary fermentable protein in pigs, assessing protein catabolic activity could provide more insight.

## 3. Impact on Microbial Catabolic Activity

Altered microbial composition in the gut due to increased protein levels will influence the catabolic activity of the community [21]. Predominant species implicated in proteolytic fermentation assessed in vitro include bacteria in the genera *Clostridium*, *Bacteroides*, *Peptostreptococci*, *Fusobacterium*, *Actinomyces Megasphaera* and *Propionibacterium* [2]. Direct plate counting results showed that different AA were favoured by different species. For example, *Clostridium* spp. show trophic utilisation of multiple AA including lysine, glycine, arginine and proline fermentation, while *Peptostreptococci* only drive tryptophan and glutamate catabolism, whereas aromatic AA metabolism is primarily performed by *Clostridium*, *Bacteroides* and *Peptostreptococci* spp. [2]. These differences in catabolic activity are determined by the presence and activity of specific enzymes in species to enable all reactions. These catabolic reactions include deamination, to produce a carboxylic acid plus ammonia, and decarboxylation, to produce an amine plus CO_2_ [50]. The deamination step can be oxidative, reductive or coupled, i.e., Stickland reaction [51]. Each AA is fermented at a different rate and yields different products [39] as different species have different AA degradation pathways [52]. A large number of taxonomically diverse bacterial species, but most certainly not all species, contain the required degradative enzymes, including members of the *Bacillus*, *Bacteroides, Bifidobacterium* and *Clostridium* genera [53]. Therefore, either omics techniques or the analysis of the end-products are relevant strategies to investigate catabolic activity of gut microbiota in vivo. Microbial omics can aid in exploring proteolytic activity of the microbiota. However, not all techniques and databases are fully optimised to enable complete and accurate assessment of the protein catabolic activity of microbiota in the gastrointestinal tract of pigs. The golden standard for analysing metabolic activity, metatranscriptomic sequencing, provides exact information on genes currently being transcribed in the gut microbial population [54,55] but still has challenges to overcome [56,57]. Therefore, this review focused on the considerable research conducted that assessed protein fermentation end-products.

An overview of the main end-products of protein fermentation from each AA, including some example microbial genera that are involved, is shown in Figure 1. Similar to our overview of microbial composition, this review provides an overview of typical end-products detected in the intestinal tract of pigs fed different levels of dietary protein (Table 2). Briefly, short-chain fatty acids (SCFA) and ammonia are the major end-products in proteolytic fermentation by microbiota. Branched-chain fatty acids (BCFA), as typical SCFA, are formed from branched-chain AA. In addition, catabolism of the sulfur-containing AA, cysteine and methionine, results in the production of hydrogen sulfide and methanethiol, respectively [58]. Biogenic amines like putrescine, agmatine, cadaverine, tyramine and histamine can be produced from ornithine and arginine, arginine, lysine, tyrosine and histidine, respectively [59]. Lastly, aromatic AA yield a series of phenolic and indolic compounds as end-products including p-cresol, indole, phenol and skatole, but the microbial metabolisation rate for aromatic AA is low compared to other AA [39].

### 3.1. Ammonia

Ammonia in the gastrointestinal tract has different origins, which can be unrelated to protein fermentation. However, more than 70% of the ammonia in the ileal digesta of pigs was generated by microbial fermentation of dietary protein and endogenous protein, while approximately 30% came from urea hydrolysis as determined using stable isotope labelled valine and urea [62]. Higher ammonia concentrations were observed to increase along the intestinal tract of piglets, being low in the stomach and high in the colon [63]. As shown in Table 2, increasing protein content in the diet of pigs resulted in increased ammonia concentrations in digesta or faeces in almost all studies. Ammonia diffuses across the intestinal barrier in large amounts, but equalled around 13 mmol/L in the distal colon of high protein-fed piglets [63]. A high ammonia concentration (20 mmol/L) was found to have harmful effects on the human colonic epithelium [64]. Potential mechanisms including interference with colonocytes metabolism, impaired barrier function and promotion of inflammatory signals were reviewed [65].

### 3.2. SCFA/BCFA

Acetate, propionate and butyrate are typical SCFA produced upon carbohydrate fermentation [66]. However, these end-products can also appear, although at relatively lower rates, as a result of AA fermentation [39]. Acetate results mainly from the fermentation of alanine, aspartate, glutamate, glycine, lysine, threonine and serine, propionate from fermentation of aspartate, alanine, threonine and methionine, whereas butyrate can be mainly formed from serine, glutamate, lysine and methionine. Unique SCFA that are only produced in proteolytic fermentation are BCFA including isobutyric acid produced from valine, isovaleric acid produced from leucine and 2-methyl-butyrate produced from isoleucine [39]. The concentration of BCFA increases from the ileum to colon in piglets as proteolytic activity of microbiota increases distally [63]. Production of BCFA can be increased rapidly by higher dietary protein level, as detected in an in vitro model of human colonic microbiota [67].

As shown in Table 2, when increasing dietary protein, increased concentrations of BCFA as well as SCFA were observed in the digesta collected from the ileum, caecum and colon as well as faeces of growing pigs [11,20,30,33,38,42,68]. However, also several studies reported unchanged SCFA and BCFA concentrations in digesta from pigs fed with high levels of dietary crude protein [19,20,32,63] and there were a few studies that observed decreased SCFA [34,41] and BCFA [34] concentrations. A possible explanation for the inconsistent results could also be related to the different levels and types of supplemented carbohydrates which might have suppressed AA fermentation [34]. The proportions of fermentable protein and carbohydrates that are available for the microbiota altered by an experimental diet needs to be more clearly defined to enable further interpretation of the results. In addition, in some studies, BCFA is reported as a proportion of SCFA, which may lead to different conclusions if the concentration of SCFA changes [24,69]. In general, increased dietary protein can result in higher SCFA and BCFA concentrations in the intestine of pigs, although data are not consistent between and within studies.

### 3.3. Biogenic Amines

Biogenic amines are mainly produced from AA decarboxylation by microbiota including species in the genera *Clostridium*, *Lactobacillus*, *Veillonella*, *Bifidobacterium* and *Bacteroides* as reviewed by Smith and Macfarlane [61]. Although these amines can be rapidly absorbed and detoxified by monoamine and diamine oxidases in the gut epithelium [70], high concentrations of amines like histamine resulting from high protein diet were associated with diarrhoea in pigs [11], likely through the induction of Cl^—^ secretion [71]. However, piglets also have the adaptive capacity for protein fermentation products, as piglets fed with high fermentable protein also had increased colonic activity of histamine-degrading enzymes [72]. The overall effect of biogenic amines is not clear since the precise functions of other amines remain largely unknown.

In general, increased dietary crude protein levels result in higher concentrations of total amines, putrescine, histamine, tyramine, cadaverine, spermidine and methylamine in digesta and faeces of pigs [11,20,30,31,34,41,42,63,68]. However, unchanged concentrations of amines in the ileum were observed in a study which also found no effects of dietary crude protein level on other metabolites such as SCFA and ammonia [20]. Furthermore, some studies detected different results in the different intestinal segments, or between the applied protein levels and sampling moments [33,46,48]. For example, the concentration of tyramine was only increased in the jejunum but not colon when dietary crude protein level was increased by 6% units [48]. Furthermore, increased concentrations of cadaverine in caecal digesta as a result of higher dietary crude protein levels were found on day 25 and 45 but not day 10 [33].Overall, in most studies, increasing the dietary crude protein level results in increased biogenic amine levels in the digesta and faeces of pigs.

### 3.4. Indolic and Phenolic Compounds

Indolic and phenolic compounds are the major metabolites of bacterial fermentation of the aromatic amino acids [60]. Indole and skatole produced from tryptophan as well as phenol produced upon fermentation of tyrosine in the large intestine can be absorbed, detoxified and excreted mainly as p-cresol. Phenol has been shown to impair colonic barrier function due to inhibition of respiration and proliferation [73], whereas indole is suggested to have beneficial effects like increased transepithelial resistance [74].

The effect of increasing dietary protein on the concentration of indolic and phenolic compounds in the intestine differed between different studies, although there was a general increase (Table 2). The concentration of phenol was increased in the caecum, colon and faeces upon increasing protein intake [41,42,63], whereas a decreased concentration was detected in the stomach and ileum [63]. Levels of skatole and indole in the intestine and faeces were also increased by feeding a high protein diet [41,42,75]. In addition, an unchanged faecal concentration of indole was also reported on the last sampling date during a long-term study [42]. As for p-cresol, an increased concentration was observed in distal colon and faeces upon increasing dietary protein levels [42,63], although decreased concentrations were also detected in another study [42].

### 3.5. Other Metabolites

Other metabolites of protein catabolism are produced in much lower levels and are rarely analysed or detected in studies. Nevertheless, these metabolites could be potentially harmful to the gastrointestinal epithelium, even at low concentrations. For instance, H_2_S, a compound that is produced from the sulfur-containing AA (cysteine and methionine), has concentration-dependent effects on both pro- and anti-inflammatory responses, smooth muscle relaxation and pro- and antinociception in the gastrointestinal system [76].

### 3.6. Overall Impact on Microbial Catabolic Activity

Overall, these findings indicate that high protein diets significantly increase the microbial fermentation of protein, peptides or AA, which was shown by the increased concentrations of metabolites derived from microbial AA metabolism, especially in the distal part of the intestine. However, the concentration of end-products in digesta or faeces does not directly reflect microbial catabolic activity as the concentration is dependent on the rate of production and disappearance by, e.g., absorption by enterocytes. In vitro studies can be useful to investigate microbial capabilities in this respect. For example, in vitro gas production techniques were conducted to investigate the fermentation of different protein sources by using pig faeces as an inoculum [77]. Batch-culture studies with human faecal microbiota also showed the metabolite profile from peptides and AA fermentation [39,60]. Although in vitro studies can indicate the fermentability of protein, the in vivo situation is more complex and multiple interactions with the host occur due to various aspects including passage, absorption and ratios between nutrients (C/N). Of the discussed metabolites, BCFA/SCFA are regarded as beneficial for intestinal health, where SCFA can act as an energy source for enterocytes and stimulate cell proliferation and differentiation [78,79]. Moreover, acids lower the luminal pH and favour growth of certain bacteria which suppress the growth of pathogens such as specific *E. coli* types [80]. Nevertheless, pH was not altered in most studies and even increased (Table 2) as higher levels of neutral and alkaline metabolites were produced during protein fermentation. Moreover, increased colonic expression of genes involved in mucosal cell turnover and proinflammatory reactions were found to be associated with high concentrations of ammonia, biogenic amines and other yet-unidentified potential toxic metabolites induced by feeding a high protein diet [11]. The effects of each specific metabolite can be studied in vivo; however, for studying the overall effect of protein fermentation, this review focused on the health effects of increasing dietary protein levels in animal studies.

## 4. Impact on the Gut and Host Health

The formation of AA-derived metabolites and their mechanistic effects in vitro and ex vivo have been reviewed by Gilbert et al. [4]. Here, we focused on in vivo studies to evaluate the overall impact of increased dietary protein on the intestine and overall health or performance of pigs.

### 4.1. Intestinal Morphology

Morphological and functional changes were reported due to increasing dietary protein intake, although only a few studies determined the effects on the large intestine (Table 3). By increasing dietary protein level, an increased relative weight of the large intestine and higher crypt depth in the colon were observed [31,81]. Piglets with diarrhoea had deeper crypts in the distal colon, but crypt depth was negatively correlated with the colonic concentration of protein fermentation products [82]. It could be that the deeper crypts were a response to butyrate, showing direct epigenetic effects on key cell-cycle transcription factor Foxo3 and regulating stem cell growth inhibition. It has therefore been proposed that crypts subsequently elongate in response to increased butyrate concentrations as a protective measure [83]. Generally, more studies looked into the effects of dietary protein on small intestinal morphology, since it is closely related to nutrient absorption [45]. As shown in Table 3, longer villi and deeper crypts in the small intestine were found as dietary protein level increased [31,38,46,47,48,84]. On the contrary, few studies found that intestinal morphology was not altered by protein level [19,22]. The difficultly of assessing the effect of protein level, and thereby the differences in fermentable protein, on morphology might be related to the different protein sources used, which can have a different effect on the small intestinal morphology [85].

### 4.2. Intestinal Barrier Function

Apart from the morphology, the intestinal barrier is a critical line of defence against pathogens, antigens or toxins [13]. This intestinal integrity, which is maintained by tight junctions between epithelial cells is, therefore, an important aspect of gut functioning [86]. Higher dietary crude protein levels resulted in greater expression of tight junction proteins in the small intestine [31]. Similarly, increased dietary protein level resulted in a higher count of mucus-containing goblet cells as well as greater gene expression of tight junction proteins like occludin, ZO-3, claudin-1 and claudin-7 in the proximal colon [31,81]. Therefore, intestinal permeability can be improved as decreased serum level of lipopolysaccharides was found [31]. Nevertheless, reduced expression of colonic claudin-1, claudin-2 and claudin-3 with unaltered barrier function was also found in the piglet colon [63]. Therefore, the large intestine of pigs was proposed to have a mucosal adaption to maintain barrier function and epithelial homeostasis. Besides, increased expression of cell turnover-related genes as well as genes related to pro- and anti- inflammatory responses were also detected in the proximal colon of pigs fed with higher protein [11]. However, these increases could be in response to increased permeability or act as a protective barrier from increased permeability. The above-mentioned studies indicate that high protein diets lead to longer villi and deeper crypts in the small intestine and increased expression of tight junction proteins. This could contribute to the lower count of intraepithelial lymphocytes in the proximal colon that was observed in piglets fed a 20% crude protein diet compared to 16% [81]. Nevertheless, increased *TNF-α, IL-1β* and *IL-6* were found in the colon of pigs fed with high-protein diets [87] or an increased NF-kB activation in the ileum, that may also lead to similar pro-inflammatory cytokines [84].

### 4.3. Diarrhoea Incidence and Growth Performance

These local effects on the intestinal tissue make it difficult to conclude whether increasing dietary protein levels, and thereby possibly protein fermentation, is detrimental for health. Therefore, the effects of fermentable protein on gut health or overall health should not be determined solely based on molecular, cellular or morphological responses but also by including clinical performance like diarrhoea incidence and growth performance of pigs.

Protein fermentation could affect faecal fluidity. For example, several studies observed that increasing dietary protein level led to decreased faecal consistency, as an indicator of diarrhoea, in pigs [11,12,32,47,88]. Reduced sodium absorption in the distal colon and, thereby, reduced water absorption, resulting from increased levels of H_2_S might also play a role [89]. Moreover, vulnerability to pathogen invasion and, thereby, diarrhoea incidence could be an underlying factor. The proliferation of pathogens like *Bacteroides* and *Clostridium* species, resulting from increased substrate availability and increased pH, was favoured (Table 1). Nevertheless, there are also studies observing unchanged faecal consistency upon increasing dietary protein [20,38,81], possibly because infection pressure was low in the relatively clean experimental environment or other factors may interfere. In enterotoxic *E. coli* challenged pigs, Heo et al. [88] estimated, based on a meta-analysis of protein level studies, that a protein restriction of 60 g/day/pig would reduce post-weaning diarrhoea incidence parallel with a declined concentration of protein fermentation products.

Even though there appears to be a clear link with faecal consistency, a low protein diet might reduce animal performance, even when limiting AA were supplemented to optimise ileal digestible AA levels [30,34]. Increased body weight [6] and average daily gain during the experimental period was observed in high protein fed animals despite higher incidence of diarrhoea [12,30,32,34,38,46,47,48]. The reduced growth in animals fed a lower protein diet, even though limiting AA were supplemented, may be explained by a different absorption kinetics between the different diets which influences the metabolic utilisation of nutrients, such as amino acid oxidation and protein deposition [90,91]. Besides, the digestibility of crude protein can be reduced when it is provided at low level with AA supplemented [42]. Meanwhile, a reduced feed conversion ratio was found as protein level increased in most of the studies (Table 3) except for the long-term study (112 days) of Yu et al. [46].

In conclusion, increasing dietary crude protein can affect small intestinal morphology, towards an increase in villus height and crypt depth. Interestingly, relatively few studies include intestinal morphology or barrier function measurements in the large intestine. An increase in dietary crude protein often results in increased diarrhoea incidence. It is difficult to determine the effect of protein fermentation as such on performance based on the studies with a difference in dietary protein level. As high protein diets may also provide additional AA (depending on the digestibility of the dietary protein sources used), it remains to be investigated whether the protein fermentation might influence an animal’s potential to thrive.

## 5. Conclusions and Perspectives

Almost all the studies reviewed above designed diets with different protein levels and, thereby, resulted in different levels of fermentable protein available for the intestinal microbiota. To maintain animal growth, some studies balanced the ileal digestible essential AA by supplementing the low protein diets with certain crystalline AA; in most studies, this was lysine, methionine, cysteine, threonine and tryptophan [30,32,46,47,84]. However, other AA like isoleucine and valine could become limiting factors for piglet performance when fed low protein diets [23]. Moreover, a reduction in the average daily feed intake of pigs in high protein groups might also occur [22]. Therefore, when studying protein fermentation by using the strategy of increased dietary protein levels, feed intake measurements should be included or feed intake should be kept equal. Furthermore, when different protein sources are used to reach a contrast in (fermentable) protein level, the measurement of ileal N flow could be important to validate whether the desired contrast was achieved.

Overall, increasing fermentable protein by increasing dietary protein level will slightly alter microbial diversity and richness and can stimulate change in microbial composition in the pig intestine, especially in young piglets. The intestinal microbiota shifts to a more N-utilising community and leads to increased protein catabolic activity, as evidenced by higher concentrations of protein-derived products like ammonia, amines and BCFA. This can be associated with increased diarrhoea incidence, although animal performance was not impaired in all cases.

To reduce protein fermentation and its potentially harmful effects, supplementing the diet with fermentable carbohydrate has been investigated. Increased bacterial utilisation of fermentable carbohydrate as an energy source in the large intestine can promote the incorporation of ammonia, dietary protein and amino acids into bacterial protein to facilitate biomass production [93,94]. Pieper et al. [11] found that dietary inclusion of more carbohydrates decreased protein fermentation products and resulted in improved faecal consistency in piglets. Therefore, changes in gut and host physiology induced by protein fermentation might be ameliorated by balancing the ratio between carbohydrates and nitrogen available for the microbiota.

## Figures and Tables

**Figure 1 microorganisms-08-01735-f001:**
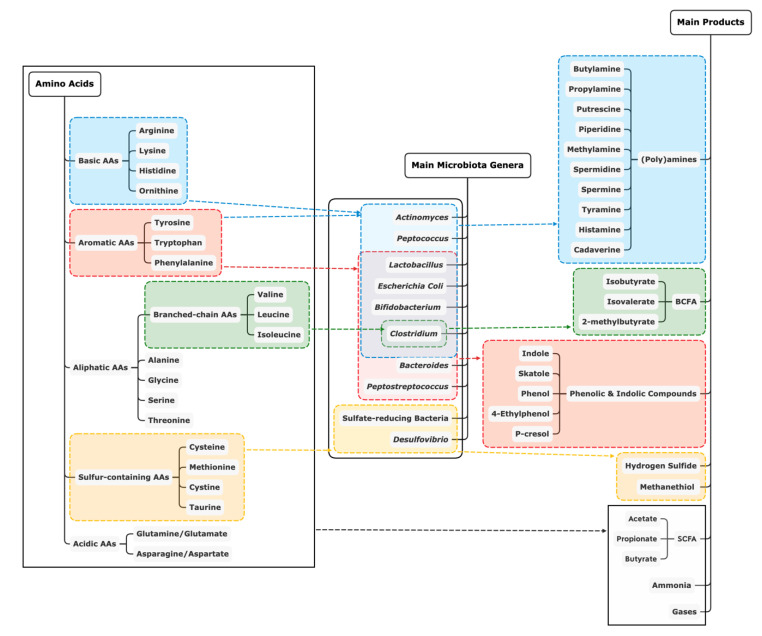
Overview of main end-products of protein fermentation by example genera of microbiota [2,39,60,61]. AA: amino acids; SCFA: short chain fatty acids; BCFA: branched chain fatty acids.

**Table 1 microorganisms-08-01735-t001:** Microbial composition shifts in response to dietary crude protein level in the intestinal tract of pigs.

Age (d)	Crude Protein Level (%)	Period (d)	Method	Microbiota Responses								Source
				Richness	Diversity	Phylum	Class	Order	Family	Genus	Species	
17	19 vs. 21	21	Culture-Based	Caecum =								[47]
17	18 vs. 23	14	Culture-based; TRFLP	-		Ileum, colon and rectum =: Total Coliforms, lactic acid producing bacteria						[32] ^a^
				Colon ↑ 7d post challenge		Colon =	=	↓: Clostridiales ↑: Unclassified Clostridiales	↓: *Lachnospiraceae*	↓: *Roseburia*	-	
18	17 vs. 19 vs. 21 vs. 23	21	Culture-Based	-	-	Ileum =: Aerobic spore formers, Anaerobic spore formers, *Enterobacteriaceae, Enterococci, E. coli*, Total Coliforms						[38]
24	16 vs. 20	21	Culture-based	-	-	Faeces =: *Enterobacteria* spp., Lactobacilli spp.						[23]
25	15 vs. 20	21-23	Real-time PCR	-	-	Proximal colon =: Total bacteria	-	-	*=: Enterobacteria*	=: Lactobacilli, *Bacteroides*	*=: Cl. coccoides* ↑: *Cl. leptum*	[11]
26	15 vs. 22	28	Culture-based	-	-	Ileum and colon =: *E. coli*, Total Coliforms, Lactobacilli						[19]
28	14 vs. 17 vs. 20	45	16S rRNA, V3-V4 regions	Jejunum and colon =		=	-	-	-	=	-	[48]
28	13 vs. 18 vs. 23	14	Culture based	-	-	Ileum = Proximal colon ↑: Coliforms ↓: *lactobacillus* to Coliforms ratios Faeces ↑: Coliforms						[49]
28/40	13 vs. 23	14	Culture based	-	-	Colon ↑: Lactobacilli =: Coliform, *Lactobacillus* to Coliforms ratio Faeces numerical ↑enterotoxigenic *E. coli*						[49]
~35	14 vs. 20	45	16S rDNA, V6–V8 regions; DGGE	-	Caecum ↑ (d25)	↑: Firmicutes (d25) =: total bacteria, Bacteroidetes	-	-	-		↑: *Clostridium* cluster IV (d25) =: *Clostridium* cluster XIVa (d10, 45)	[33]
35	10 vs. 13 vs. 16	112	16S rRNA, V3–V4 regions	Colon =		-	-	-	-	↑: *Streptococcus, Lactobacillus, Turicibacter* ↓: *Prevotella, Lachnospira Ruminococcus, Dorea, Candidatus,* Unclassified *Clostridiales,* Uncultured *Peptococcaceae*	-	[46] ^b^
42	14 vs. 18	143	DGGE; Real-time PCR; 16S rRNA, V6–V8 regions	Caecum = Faeces		Caecum =: total bacteria, Firmicutes ↓: Bacteroidetes Faeces =: total bacteria, Firmicutes, Bacteroidetes	-	-	-	-	Caecum ↑: *E. coli* *(d77)* ↓: *Clostridium* cluster IV, *Clostridium* cluster XIVa Faeces ↑: *E. coli (d77)* =: *Clostridium* cluster IV, *Clostridium* cluster XIVa	[41,42] ^b^
45	14 vs. 15 vs. 17 vs. 20	28	Real-time PCR	-	Ileum, caecum = Colon: ↑ ↓=	Ileum, caecum, colon =: total bacteria, Firmicutes, Bacteroidetes	-	-	-	Caecum ↑: *Bifidobacterium* (15, 20%) Colon ↑: *Bifidobacterium* Ileum, caecum, colon =: *Lactobacillus*	Colon↓: *E. coli* (15%) Ileum, caecum, colon =: *Clostridium* cluster IV, *Clostridium* cluster XIVa	[34]
70	13 vs. 16	100	16S rRNA, V1-V3 regions	Caecum, colon =		=	-	-	Caecum ↓: *unclassified Peptostreptococcaceae, uncultured Lachnospiraceae, and uncultured Erysipelotrichaceae* Colon ↓: *Unclassified Clostridiaceae, and Erysipelotrichaceae incertae sedis*	Caecum ↑: *Lactobacillus* ↓: *Prevotella, Coprococcus* Colon ↑: *Streptococcus* ↓: *Sarcina, Peptostreptococcaceae incertae sedis, Mogibacterium, Subdoligranulum, Coprococcus*	-	[40] ^b^
~80	12 vs. 15 vs. 18	30	16S rRNA, V3–V4 regions	Ileum, colon =		Ileum ↑↓: Firmicutes ↓↑: Proteobacteria Colon ↓↑: Bacteroides ↑↓: Firmicutes, Spirochaetae ↑: Verrucomicrobia	-	-	Ileum ↓↑: *Streptococcaceae, Enterobacteriaceae, Leuconostocaceae* ↑↓: *Lactobacillaceae, Closridiaceae_1, Micrococcaceae* Colon = ↓: *Lachnospiraceae,* ↑↓: *Veillonellaceae* ↑: *Ruminococcaceae*	Ileum ↓↑: *Streptococcus, Escherichia-Shigella, Weissella* ↑↓: *Lactobacillus, Clostridium_sensu_stricto_1* Colon ↓↑: *Ruminococcaceae_UCG- 005, Norank_f _Bacteroidales_S24-7_group* ↑↓: *Streptococcus* ↑: *Prevotellaceae_NK3B31_group*	-	[31]
Finishing	10 vs. 13 vs. 16	50	16S RNA, V3–V4 regions	Ileum, Colon =		Ileum =↑: Firmicutes =↓: Proteobacteria *=: Actinobacteria* Colon=: Firmicutes, Bacteroidetes, Spirochaetae	-	-	Ileum =↑: *Clostridiaceae_1* =↓: *Enterobacteria* Colon ↓: *Clostridiaceae_1, Erysipelotrichaceae* =↑: *Rikenellaceae* ↑↓: *Peptostreptococcaceae* *=: Spirochaetaceae*	Ileum = ↑: *Clostridium_sensu_stricto_1* ↓: *Escherichia-Shigella* Colon =↓: *Clostridium_sensu_stricto_1, Turicibacter*	-	[30]
Finishing	15 vs. 20	~30	Culture-based	-	-	-	-	-	-	-	Caecum = Colon ↓: *Bifidobacteria* spp. =: Lactobacilli spp., *Enterobacteria* spp.	[24]

=: no effect; -: not determined in the study; ↑ or ↓: increased or decreased result in pigs fed with high protein levels compared to low levels; ↑↓or ↓↑: differences among low-moderate-high protein treatment groups; = ↑: no effect was observed between the first two protein levels, but from the lowest to the highest level, there was an increase; = ↓: no effect was observed between the first two levels, but from the lowest to the highest level, there was a decrease; a: all pigs were challenged with enterotoxigenic *Escherichia coli*, b: long-term studies and protein level changed over time based on nutritional requirement; therefore, only the final protein level was indicated; d: days; vs.: versus; TRFLP: terminal restriction fragment length polymorphism; PCR: polymerase chain reaction; RNA: ribonucleic acid; rRNA: ribosomal ribonucleic acid; DGGE: denaturing gradient gel electrophoresis.

**Table 2 microorganisms-08-01735-t002:** Catabolic shifts in response to dietary crude protein level in the intestinal tract of pigs.

Age (d)	Duration (d)	Crude Protein Level (%)	Segment	Change in Digesta Concentrations					Source
				pH	Short-Chain Fatty Acids	Branched-Chain Fatty Acids	Biogenic Amines/Indolic and Phenolic Compounds	Ammonia	
17	14	18 vs. 23	Colon	-	=	=	-	↑	[32] ^a^
17	21	19 vs. 21	Caecum	-	=	-	-	↑	[47]
18	21	17 vs. 19 vs. 21 vs. 23	Duodenum	=	↑: propionic, valeric	=	-	↑	[38]
			Jejunum	=	=	=	-	↑	
			Ileum	↑	↑: acetic, propionic, valeric	↑: isobutyric, isovaleric	-	↑	
21	28	18 vs. 26	Faeces	-	↑	= (%)	-	↑	[69] ^a^
21	21	15 vs. 19	Colon	-	↑	=	-	-	[81]
24	21	16 vs. 20	Faeces	=	↑: total SCFA, butyric acid (%)	=	-	-	[23]
25	21-23	15 vs. 20	Colon	-	↑	↑	↑: putrescine, histamine, spermidine	↑	[11]
25	21	18 vs. 26	Stomach	=	=	-	↓: cadaverine, phenol	↑	[63]
			Ileum	=	=: total SCFA ↑: acetate (%), butyrate (%), propionate (%)	-	↑: total amines, putrescine, ↓: phenol	↑	
			Caecum	=	↑: total SCFA, acetate, butyrate, propionate (%), butyrate (%)	=	↑: histamine, 4-ethylphenol	↑	
			Proximal colon	=	=	=	↑: total amines	↑	
			Distal colon	=	=: total SCFA ↑: butyrate ↓: acetate (%)	=	↑: cadaverine, phenol, p-cresol, skatole	=	
25	22	18 vs. 26	Proximal colon	-	↑: total SCFA, butyrate	↑	↑: putrescine, histamine, and spermidine	↑	[68]
26	28	15 vs. 22	Ileum	-	=	=	-	↑	[19]
			Colon		=	=	-	=	
28	14	20 vs. 24	Caecum	=	↑: acetic acid	↑: isobutyric acid, isovaleric acid	↑: putrescine	↑	[20]
28	45	14 vs. 17 vs. 20	Jejunum	-	=	=	= ↑: tyramine	=	[48]
			Colon	-	=	=	=	= ↑	
33	14	20 vs. 24	Ileum	=	=	=	=	=	[20]
~35	10	14 vs. 20	Caecum	-	=	=	=	=	[33]
	25				↑: acetate =: total SCFA	↑	↑: cadaverine	↑	
	45				↑: acetate, total SCFA	↑	↑: cadaverine	↑	
35	112	10 vs. 13	Colon	-	=	↑: isobutyrate, isovalerate	=	=	[46] ^c^
		13 vs. 16		-	↑: total SCFA, acetate	=	↑: tryptamine, putrescine, cadaverine	↑	
42	78	10 vs. 14	Caecum	-	↓: acetate, propionate	↑: isobutyrate	↑: tyramine, cadaverine, phenol and indole	↑	[41] ^b,c^
	35		Faeces	-	=: total SCFA, acetate, propionate, butyrate, valerate	=	↑: total amines, methylamine, tryptamine, cadaverine, tyramine, skatole, p-cresol, indole =: putrescine, spermidine, spermine, phenol	↑	[42] ^b^
	143				↑: total SCFA, acetate, propionate =: butyrate, valerate	↑: isobutyrate, isovalerate	↑: total amines, putrescine, spermidine, methylamine, tryptamine, cadaverine, tyramine, skatole, phenol ↓: p-cresol =: spermine, indole		
45	28	14 vs. 15 vs. 17 vs. 20	Ileum	-	↓: total SCFA, butyrate ↑=: Valerate	↓: isobutyrate ↑: isovalerate = BCFA	↑: total amines, cadaverine, putrescine	↑	[34]
			Caecum	-	↑: total SCFA, butyrate	↓: isovalerate, BCFA	↑: histamine, spermidine	↑	
			Colon	-	↑: total SCFA, acetate, valerate	=	↑: total amines, tryptamine, putrescine, spermidine	↑	
70	100	15/13 vs. 18/16	Caecum	-	=: total SCFA, acetate, propionate, butyrate, valerate	↑: isobutyrate, isovalerate; BCFA/SCFA	-	-	[40] ^c^
			Colon	-	=	=	-	-	
~80	30	12 vs. 15 vs. 18	Ileum	-	=	=	↑: putrescine, histamine, spermidine	-	[31]
			colon	-	↑: acetic acid	=	↑: cadaverine, spermidine	-	
Finishing	50	10 vs. 13 vs. 16	Ileum	-	↑: acetic acid ↓: valeric acid	=	↑: methylamine, cadaverine, putrescine, histamine, spermidine	-	[30]
			Colon		↑: acetic acid, propionic acid, butyric acid	↑: isobutyric acid, isovaleric acid	↑: methylamine, putrescine, histamine, spermidine	-	
Finishing	~30	15 vs. 20	Caecum	=	=	= (%)	-	-	[24]
			Colon	=	=: total SCFA ↑: butyric acid (%)	= (%)	-	-	
			Faeces	↑	-	-	-	↑	

=: no effect; -: not determined in the study; ↑ or ↓: increased or decreased result in pigs fed with high protein levels compared to low levels; = ↑: no effect was observed between the first two levels, but from the lowest to the highest level, there was an increase; a: animal challenged with pathogen (*E. coli*); b: animal treated with antibiotics (growth promotor); c: long-term studies and protein level changed over time based on nutriment requirement. d: days; vs.: versus; SCFA: short-chain fatty acids; BCFA: branched-chain fatty acids.

**Table 3 microorganisms-08-01735-t003:** Impact of dietary crude protein level on the intestine and health of pigs.

Age (d)	Duration (d)	Crude Protein Level (%)	Growth Performance	Faecal Fluidity	Organ Weight	Intestinal Responses			Source
						Morphology	Integrity	Immunity	
17	14	18 vs. 23	↑: ADG ↓: FCR (Before)	↑ (After)	-	Ileum ↑: CD (before); ↓: VH (after), VCR	-	-	[32] ^a^
17	21	19 vs. 21	↑: ADG ↓: FCR	↑	-	Duodenum, ileum ↑: CD	-	-	[47]
18	21	17 vs. 19 vs. 21 vs. 23	↑: ADG, ADFI, BW ↓: FCR	=	=: spleen ↑: ST, liver	Duodenum = Jejunum ↑: VH, CD ↓: VCR	-	-	[38]
18	14	17 vs. 19 vs. 21 vs. 22	=: ADFI ↑: ADG ↓: FCR	↑	-	Duodenum, jejunum ↑: VH Ileum =	-	-	[92]
25	21/23	15 vs. 20	=	-	-	-	Proximal colon ↑: PCNA,	Proximal colon ↑: IL1β, IL10, TGFβ, MUC1, MUC2 and MUC20	[11]
26	28	15 vs. 22	=	-	-	= VH, CD	=: density of gut wall muscularis, serosa, mucosa	-	[19]
29	14	13 vs. 18 vs. 23	↑: ADG, =: ADFI ↓: FCR	↑	↑: ST, SI	-	-	-	[12]
28/40	14	13 vs. 23	↓: ADFI, FCR	↑	-	= VH, CD	-	-	[49] ^a^
31	45	14 vs. 17 vs. 20	↑: BW, ADG, ADFI ↓: FCR	-	-	Duodenum↑: VH, CD, VCR Jejunum↑: VH, VCR Ileum =	-	-	[48]
35	112	10 vs. 13 vs. 16	↑: BW, ADG, ADFI, FCR	-	-	Duodenum↑: VH, CD; =: VCR Jejunum↑: VCR Ileum =	-	-	[46] ^b^
35	21	16 vs. 20	=	=	↑: ST, LT (%BW) =: liver (%BW)	-	Proximal colon ↑: goblet cells	Proximal colon ↓: intraepithelial lymphocytes	[81]
45	28	14 vs. 15 vs. 17 vs. 20	=: ADFI ↑: BW, ADG ↓: FCR	-	=: heart, spleen, kidney ↑: liver, pancreas	Duodenum ↑: VH, CD Jejunum, ileum ↑: CD; ↓: VCR	-	Ileum ↑ mRNA: TLR-4, MyD88, NF-kB; ↓: TOLLIP Plasma ↑: CD3^+^T cells, IgG =: CD3^+^CD4^+^T cells, CD3^+^CD8^+^T cells	[84]
~80	30	12 vs. 15 vs. 18	-	-	-	Ileum ↑: VH (18%), CD (15%) colon ↑: CD (15, 18%)	Serum ↓ =: LPS Ileum ↑: claudin-3 (18%), claudin-7 (15, 18%)Colon ↑: occludin, ZO-3, claudin-1, claudin-7	-	[31]
Finishing	50	10 vs. 13 vs. 16	↑: BW	-	-	Ileum ↑: VH ↓: CD	Ileum ↑: claudin-1, occludin (13%) Colon =: tight junction proteins	-	[30]

↑ or ↓: increased or decreased result in pigs fed with high protein levels compared to low levels; =: no effect; ↓=: from the first to the second level, there was a decrease but no effect was observed between the second and the third level; ADFI: average daily feed intake; BW: body weight; ADG: average daily gain; FCR: feed conversion ratio; VH: villous height; CD: crypt depth; VCR: the ratio of villous height to crypt depth; ST: stomach; SI: small intestine; LI: large intestine; LPS: lipopolysaccharides; ZO-3: zonula occludens protein 3; PCNA: proliferating cell nuclear antigen; IL1β: interleukin 1 beta; IL10: interleukin 10; TGFβ: transforming growth factor beta; MUC: mucin; mRNA: messenger ribonucleic acid; TLR-4: toll-like receptor 4; MyD88: myeloid differentiation factor 88; NF-kB: nuclear factor kappa B; TOLLIP: toll-interacting protein; CD3^+^: cluster of differentiation 3; CD4^+^: cluster of differentiation 4; CD8^+^: cluster of differentiation 8; IgG: immunoglobulin G; d: days; vs.: versus; a: animal challenged with pathogen (*E. coli*); b long-term studies and protein level changed over time based on nutriment requirement.

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
