# Peer review of "Impact of Fermentable Protein, by Feeding High Protein Diets, on Microbial Composition, Microbial Catabolic Activity, Gut Health and beyond in Pigs"

_microorganisms, 2020, doi:10.3390/microorganisms8111735_

Round 1
Reviewer 1 Report
The protein content of piglets' diets has been linked to disease and is one of the items that farmers pay attention to when feeding.
A review article on the relationship between protein and diarrhea in piglets will be of interest to farmers as well as researchers.
However, this manuscript needs to be revised and the following questions answered
Why does protein content seem to change from low to high concentrations in all Tables?
Normally, the protein content of piglets' diet changes from higher to lower concentrations as they grow.
The style of the Table must be consistent.
The number of References is not the same. The number is 96 in the text, but 97 are listed.
Based on the above, this manuscript should be properly proofread.
Author Response
Response to Reviewer 1 Comments
Point 1: Why does protein content seem to change from low to high concentrations in all Tables? Normally, the protein content of piglets' diet changes from higher to lower concentrations as they grow.
Response 1: The protein level in the tables means the protein level fed to the different treatment groups. All studies included had a parallel design with two (or more) groups of pigs fed with high or low protein during same experimental period. The studies had different research questions, varying from studying the effect of increasing protein level or studying the effect of feeding a low protein level. In the table, we decided to visualise the different protein levels applied consistently, by always comparing increased dietary protein level against the low protein level as ‘control’ because this would have the least protein fermentation. In long-term studies, indeed as pigs grow the protein content decreases coherently with the requirements and this was indicated by a “b” superscript in table 1 (line 287) and table3 (line 796) and a “c” superscript in table 2 (line 612). In this case, only the dietary protein levels at the end of the treatment period were shown in the column of “crude protein level”.
In order to clarify this matter in the manuscript, this is mentioned in the text in line 62-64:“The current review focuses on the effects of protein fermentation in vivo by comparing microbial composition, the formation of metabolites, and gut health between pigs fed with increased dietary protein levels compared to pigs fed with lower protein levels.” Moreover, the arrows were replaced by ‘vs.’ in all tables and the direction of results was indicated in the table legends (in line 285-287 & 611-612 &790-791).
Point 2: The style of the Table must be consistent.
Response 2: Thank you for this comment, indeed it is our intention to have consistent styles across the manuscript. This was adapted to be consistent and adherent to the journal’s format.
Point 3: The number of References is not the same. The number is 96 in the text, but 97 are listed.
Response 3: The error of “reference 97” was removed from the revised manuscript.
Reviewer 2 Report
Manuscript Number: Microorganisms-967662
Title: Impact of fermentable protein, by feeding high protein diets, on microbial composition, microbial catabolic activity and gut health and beyond in pigs
General Comment.
In this review Authors discussed in vivo effects of protein fermentation on the microbial composition and their protein catabolic activity as well as gut and overall health. This would seem to be an original contribution containing clear new aspects. On the whole it is clear and well written. The interpretations and conclusions are consistent with the data presented. Some tables are not informative, because all acronyms should be explained in their legends. All references diverge from the Journal style guidelines and require some attention.
However, the manuscript deserves some ameliorations in order to be published.
Additional points.
Line 109: “…Peng et al. 34 showed ...” instead of “…Peng et al. (2017) 34 showed…”.
Line 118: “Wellock et al. 12 detected...” instead of “…Wellock et al. (2006) 12 detected…”.
Line 130: “…in the study of Louet et al. 33 was only...” instead of “…in the study of 33 was only…”
Lines 153-154: “…of growing pigs 46.” instead of “… of growing pigs (Yu, Zhu, and Hang 2019).”
Table 1: Delete at the end of the table “181 182 183 184 185 186 187 188 189 190 191”.
Table 1: All acronyms (i.e. TRFLP, PCR, DGGE) should be explained in the legend.
Line 283: “…as reviewed by Smith and Macfarlane 61.” instead of “…as reviewed by 61.”
Line 324-346: Move these sentences in another point of the review, because they do not regard this paragraph (3.4. Indolic and phenolic compounds).
Table 2: Delete at the end of the table “349 350 351 352 353 354”.
Line 360: “…by Gilbert et al. 4.” instead of “…by Gilbert et al. (2018) 4.”
Line 414: “…Heo et al. 90 estimated…” instead of “… Heo et al. (2015) estimated…”.
Line 414: “…of Yu et al. 46.” instead of “… of Yu et al. (2019)46.”.
Table 3: Some acronyms (i.e. PCNA, IL1β, IL10, TGFβ, MUC1, etc.) should be explained in the legend.
Line 462: “…Pieper et al. 11 found…” instead of “… Pieper et al. found…”.
All references diverge from the Journal style guidelines and require some attention. For example:
Line 476: “Pharmacol. Res. 2013, 68, 95 –107.” instead of “Pharmacol Res 2013, 68 (1), 95-107.”
Lines 478-479: “FEMS Microbiol. Ecol. 1998, 25, 355–368.” instead of “FEMS Microbiology 478 Ecology 1998, 25 (4), 355-368.
Lines 481-740: etc….
Line 740: delete “97.”
Author Response
Response to Reviewer 2 Comments
Point 1: Some tables are not informative, because all acronyms should be explained in their legends.
Table 1: All acronyms (i.e. TRFLP, PCR, DGGE) should be explained in the legend.
Table 3: Some acronyms (i.e. PCNA, IL1β, IL10, TGFβ, MUC1, etc.) should be explained in the legend.
Response 1: All acronyms are explained in line 288-290 ‘TRFLP: terminal restriction fragment length polymorphism; PCR: polymerase chain reaction; RNA: ribonucleic acid; rRNA: ribosomal ribonucleic acid; DGGE: denaturing gradient gel electrophoresis’ and line 790-795 ‘ADFI: average daily feed intake; BW: body weight; ADG: average daily gain; FCR: feed conversion ratio; VH: villous height; CD: crypt depth; VCR: the ratio of villous height to crypt depth; ST: stomach; SI: small intestine; LI: large intestine; LPS: lipopolysaccharides; ZO-3: zonula occludens protein 3; PCNA: proliferating cell nuclear antigen; IL1β: interleukin 1 beta; IL10: interleukin 10; TGFβ: transforming growth factor beta; MUC: mucin; mRNA: messenger ribonucleic acid; TLR-4: toll-like receptor 4; MyD88: myeloid differentiation factor 88; NF-kB: nuclear factor kappa B; TOLLIP: toll-interacting protein; CD3+: cluster of differentiation 3; CD4+: cluster of differentiation 4; CD8+: cluster of differentiation 8; IgG: immunoglobulin G; d: days; vs.: versus.’
Point 2: Line 109: “…Peng et al. 34 showed ...” instead of “…Peng et al. (2017) 34 showed…”.
Line 118: “Wellock et al. 12 detected...” instead of “…Wellock et al. (2006) 12 detected…”.
Line 130: “…in the study of Lou et al. 33 was only...” instead of “…in the study of 33 was only…”
Lines 153-154: “…of growing pigs 46.” instead of “… of growing pigs (Yu, Zhu, and Hang 2019).”
Table 1: Delete at the end of the table “181 182 183 184 185 186 187 188 189 190 191”.
Line 286: “…as reviewed by Smith and Macfarlane 61.” instead of “…as reviewed by 61.”
Table 2: Delete at the end of the table “349 350 351 352 353 354”.
Line 360: “…by Gilbert et al. 4.” instead of “…by Gilbert et al. (2018) 4.”
Line 414: “…Heo et al. 90 estimated…” instead of “… Heo et al. (2015) estimated…”.
Line 414: “…of Yu et al. 46.” instead of “… of Yu et al. (2019)46.”.
Line 462: “…Pieper et al. 11 found…” instead of “… Pieper et al. found…”.
Line 740: delete “97.”
Response 2: All changes were made according to suggestions.
Line 118: “…Peng et al. 34 showed ...” instead of “…Peng et al. (2017) 34 showed…”.
Line 127: “Wellock et al. 12 detected...” instead of “…Wellock et al. (2006) 12 detected…”.
Line 140: “…in the study of Lou et al. 33 was only...” instead of “…in the study of 33 was only…”
Lines 165: “…of growing pigs 46.” instead of “… of growing pigs (Yu, Zhu, and Hang 2019).”
Table 1: Deleted at the end of the table “285-305”.
Line 407-408: “…as reviewed by Smith and Macfarlane 61.” instead of “…as reviewed by 61.”
Table 2: Delete at the end of the table “611-615”.
Line 626: “…by Gilbert et al. 4.” instead of “…by Gilbert et al. (2018) 4.”
Line 692: “…Heo et al. 90 estimated…” instead of “… Heo et al. (2015) estimated…”.
Line 704: “…of Yu et al. 46.” instead of “… of Yu et al. (2019)46.”.
Line 850: “…Pieper et al. 11 found…” instead of “… Pieper et al. found…”.
Line 1149: delete “97.”
Point 3: Line 324-346: Move these sentences in another point of the review, because they do not regard this paragraph (3.4. Indolic and phenolic compounds).
Response 3: Thank you for this comment. Indeed, the paragraph (now line 453-475) starting with “Overall, these findings…” reflects on all metabolites. A header ‘3.6 Overall impact on microbial catabolic activity’ was inserted to make this difference clear to our readers (Line 452). Additionally, also a header was inserted for the other metabolites that were discussed, namely ‘3.5 Other metabolites’ (Line 445).
Point 4: All references diverge from the Journal style guidelines and require some attention. For example:
Line 476: “Pharmacol. Res. 2013, 68, 95 –107.” instead of “Pharmacol Res 2013, 68 (1), 95-107.”
Lines 478-479: “FEMS Microbiol. Ecol. 1998, 25, 355–368.” instead of “FEMS Microbiology 478 Ecology 1998, 25 (4), 355-368.
Lines 481-740: etc….
Response 4: A wrong-cited reference (69) was deleted from the text (in line 396) and reference list. References after 69 were re-ordered. Format of all reference was changed according the Journal style guidelines.
Round 2
Reviewer 1 Report
The authors responded to this reviewer's comments.
Tables are now easier to read.